# Beyond Classification: Whole Slide Tissue Histopathology Analysis By End-To-End Part Learning

**Chensu Xie**[1,2]                                         XIC3001@MED.CORNELL.EDU
**Hassan Muhammad**[1,2]                              HAM2024@MED.CORNELL.EDU
**Chad M. Vanderbilt**[2]                                   VANDERBC@MSKCC.ORG
**Raul Caso**[2]                                               CASOJRR@MSKCC.ORG
**Dig Vijay Kumar Yarlagadda**[2]                          YARLAGAD@MSKCC.ORG
**Gabriele Campanella**[1,2]                                CAMPANEG@MSKCC.ORG
**Thomas J. Fuchs**[1,2]                                      FUCHST@MSKCC.ORG

[1] *Weill Cornell Graduate School of Medical Sciences, Cornell University, New York, USA*

[2] *Department of Pathology, Memorial Sloan Kettering Cancer Center, New York, USA*

## Abstract

An emerging technology in cancer care and research is the use of histopathology whole slide images (WSI). Leveraging computation methods to aid in WSI assessment poses unique challenges. WSIs, being extremely high resolution giga-pixel images, cannot be directly processed by convolutional neural networks (CNN) due to huge computational cost. For this reason, state-of-the-art methods for WSI analysis adopt a two-stage approach where the training of a tile encoder is decoupled from the tile aggregation. This results in a trade-off between learning diverse and discriminative features. In contrast, we propose end-to-end part learning (EPL) which is able to learn diverse features while ensuring that learned features are discriminative. Each WSI is modeled as consisting of $k$ groups of tiles with similar features, defined as parts. A loss with respect to the slide label is backpropagated through an integrated CNN model to $k$ input tiles that are used to represent each part. Our experiments show that EPL is capable of clinical grade prediction of prostate and basal cell carcinoma. Further, we show that diverse discriminative features produced by EPL succeeds in multi-label classification of lung cancer architectural subtypes. Beyond classification, our method provides rich information of slides for high quality clinical decision support.

## 1. Introduction

Histopathology analysis is a fundamental step in the pipeline of cancer care and research which includes diagnosis, prognosis, treatment selection, and subtyping. Recent years saw an increase in the digitization of histopathology whole slide images (WSI) and development of computational methods for WSI analysis. WSIs, being extremely high resolution giga-pixel images, face various challenges on their analysis. Inputting a WSI at highest resolution directly through a convolutional neural network (CNN) for training is impossible due to huge computational cost. To match traditional input sizes for traditional feed-forward CNN models, a typical WSI would need to be down-sampled by a factor of $\sim 100\times$, resulting in the loss of cellular and structural details which are critical for prediction.

To overcome this bottleneck, state-of-the-art methods for WSI analysis adopt a two-stage approach. First, by training a CNN model on small image tiles sampled from WSIs at high resolution, each tile is encoded to a prediction score or a feature vector of low dimension. Second, an aggregation model is learned to integrate the obtained tile level information for whole slide prediction. For the first-stage tile encoder training, early works utilized extra supervision from pathologists beyond slide labels (Mousavi et al., 2015; Cruz-Roa et al., 2014). These annotations are usually extensive, pixel-wise labels, which are difficult to obtain for large datasets due to the time-consuming process for pathologists who already have high workload on service.

Conclusively, Weakly-supervised approaches optimized for simple slide level labels are of high interest. One track of weakly-supervised methods utilize unsupervised techniques (Muhammad et al., 2019; Tellez et al., 2019; Wang et al., 2019; Zhu et al., 2017b; Vu et al., 2015) to learn a tile encoder, and only supervise the training of the aggregation model with slide label. The other track of these weakly-supervised methods adopt the multiple instance learning (MIL) framework which assumes that the binary slide label is represented by the existence of positive tiles, such as tumor versus non-tumor classification (Campanella et al., 2019; Yao et al., 2019; Chen et al., 2019; Zhu et al., 2017b,a; Xu et al., 2017; Hou et al., 2016; Xu et al., 2014a,b; Cosatto et al., 2013). By training the tile encoder with tiles selected based on their score, MIL approaches aim to identify the most discriminative tiles for cancer classification, while the unsupervised approaches tend to incorporate diverse tile groups across the entire dataset for more complicated tasks such as survival analysis (Muhammad et al., 2019; Zhu et al., 2017b).

In contrast to the aforementioned two-stage approaches, we proposed to impose strong supervision on WSIs in an end-to-end manner by modeling each WSI as consisting of $k$ groups of tiles with similar features, defined as parts. The loss with respect to the slide label is backpropagated directly through an integrated CNN model to $k$ input tiles, representing each part (Section 3.2). To the best of our knowledge, this is the first model which performs WSI analysis using end-to-end learning with an integrated CNN model. We name this method End-to-end Part Learning (EPL). We show that EPL is capable of clinical grade classification of prostate and basal cell carcinoma (Section 4.1), and the diverse discriminative features learned by it are capable of multi-label prediction of lung cancer architectural subtypes (Section 4.2). Beyond WSI classification, EPL can also provide rich information about WSIs for high quality clinical decision support, such as tissue type localization (cf. Figure 5) and region importance scoring (cf. Figure 3 heatmap bar). In addition, we theoretically consider EPL to be applicable to survival regression, treatment recommendation, or any other learnable WSI label predictions.

## 2. Related Works

### 2.1. Multiple Instance Learning For WSI Classification

Multiple instance learning approaches are popular for weakly-supervised cancer classification (Campanella et al., 2019; Yao et al., 2019; Chen et al., 2019; Zhu et al., 2017b,a; Xu et al., 2017; Hou et al., 2016; Xu et al., 2014a,b; Cosatto et al., 2013), in which scenario the binary slide label is represented by the existence of positive tiles (Campanella et al., 2019; Xu et al., 2017, 2014a,b). Since this assumption does not hold for multi-class cancer type

prediction, Hou et al. (2016) and Chen et al. (2019) applied a generalized MIL approach that selects training tiles based on their class-specific prediction score. Ilse et al. (2018) proposed classifying tissue images by learning the model attention to all tiles followed by weighted averaging of the tile features. Unfortunately, this method has to be combined with other tile sampling techniques to be feasible on WSIs.

## 2.2. Unsupervised Tile Encoding For WSI Analysis

MIL approaches iteratively select training tiles by regarding their scores as the predictivity, and use the tile scores to vote the slide label. They work perfectly in the scenario of cancer classification, but also only applicable to the problems with this assumption. Unsupervised learning of the tile encoder with constraints can produce a collection of diverse information over a WSI for more complicated targets. Muhammad et al. (2019) and Zhu et al. (2017b) learned tile features by patch reconstruction and PCA of raw image signals respectively, constructing clusters of tiles for survival analysis. Vu et al. (2015) also proposed to learn a visual dictionary of tile features for histopathological image classification. Tellez et al. (2019) drastically compressed WSIs by replacing tiles with their features from contrastive learning in position before classifying the compressed slides. These methods however decoupled the training of tile encoder and aggregation model, as the slide labels were never used in the tile encoder training. Our propose method EPL was motivated for learning diverse tile clusters in an end-to-end supervised way for complicated WSI analysis tasks.

## 3. Method

### 3.1. Two-Stage Methods For WSI Classification

As WSIs are too large to be trained by a CNN end-to-end, two-stage methods tend to compress a slide to a latent variable in low dimensional space by sampling tiles and passing them through a tile encoder $\theta_e : X \to Z$, to later predict the slide label by optimizing an aggregation model $\theta_a : Z \to Y$. Given extra tile-level annotations, Z is usually the ground truth score of sampled tiles $X$. However, for weakly supervised WSI classification with only slide level labels, learning $Z$ is not trivial.

Methods adopting unsupervised techniques to learn $Z$ decouple the training of $\theta_a$ and $\theta_e$; i.e. the slide level supervision $(Y)$ was never used for $\theta_e$ training. MIL approaches model $Z = Y$ as "pseudo labels" for tile encoder training and iteratively select predictive tiles by thresholding $Z$. Since $Z = Y$, this expectation-maximization approach can learn more discriminative features but incorporates noise during the tile encoder training. However, in more complicated tasks, such as survival regression, $Z = Y$ does not hold for any single tile; i.e. none of the single tiles is capable of telling if the patient can survive longer.

### 3.2. WSI Analysis by End-To-End Part Learning

Ideally, WSI prediction should be learned by end-to-end optimization of all parameters including $\theta_a$ and $\theta_e$ in an integrated CNN model based on all tiles of each slide. Formally,

$$maximize \quad P(Y|\theta_a, \theta_e, X) \tag{1}$$

As using all tiles creates a huge computing graph, we propose representing each WSI by $k$ groups of tiles, each called a "part" of the slide. Since the tissue connectivity varies largely across slides (needle biopsies, large/small excisions, etc.), these parts are not defined spatially but by feature similarity. Tiles of a slide $S^i$ are mapped to $k$ global centroids $\{z_1, ..., z_k\}$ of tile features $\theta_e(X)$ of the whole training dataset $\{S^1, S^2, ...\}$. Then the slide specific feature centroids $\{z_1^i, ..., z_k^i\}$ are used to represent the $k$ parts of $S^i$ and connected to the aggregation module:

$$maximize \quad P(Y|\theta_a, Z), \text{ where } Z = \{z_1^i, ..., z_k^i\},\ z_k^i = 1/N \sum_{n=1}^{N} \theta_e(x_{k,n}^i) \tag{2}$$

Although the size of $\theta_a$ has been reduced largely by only taking in $k$ centroid features, the centroids still need to be computed through a huge graph by averaging all tile features of each part. To relax the problem, we approximate each centroid by randomly sampling one of its $p$ nearest tiles, while maximizing the likelihood that the encoded feature of the sampled tile is equal to the corresponding centroid. By centroid approximation, the problem is reduced to a remarkably smaller model:

$$maximize \quad P(Y|\theta_a, \theta_e, \{x_1^i, ..., x_k^i\}) + P(Z|\theta_e, \{x_1^i, ..., x_k^i\}) \tag{3}$$

Note that now the whole model can be optimized directly from end-to-end to learn discriminative features for prediction of $Y$, while defining $k$ parts with diverse features for a better representation and understanding of the whole slides. Figure 1 is an illustration of the proposed end-to-end part learning model.

### 3.3. Manifold Initialization and Training Iterations

To initialize the data manifold, tiles of the whole training dataset are randomly split into $k$ groups at the beginning. Parameters of $\theta_e$ are also initialized randomly. Then we calculate the initial global centroids $\{z_1, ..., z_k\}^0$ by $z_k = 1/N \sum_{n=1}^{N} \theta_e(x_{k,n})$ (cf. Figure 1 a). To save computing time, we only subsample 10% of training slides and 100 tiles per slide for each epoch. For a training slide $S^i$, its 100 tiles $X^i$ are assigned to the $k$ clusters according to their distance to the global centroids in feature space (cf. Figure 1 b). The average feature of the tiles from $S^i$ (cf. dashed regions in Figure 1) in each cluster is the slide specific centroid for that part: $z_k^i = 1/N \sum_{n=1}^{N} \theta_e(x_{k,n}^i)$. To represent each part, we calculate the $p$ nearest tiles to its centroid $z_k^i$ and randomly select one of them as the centroid approximation tile $x_k^i$ (cf. crosses with black outline in Figure 1). The $k$ centroid approximation tiles are then fed through $\theta_e$ followed by an aggregation layer $\theta_a$ for slide label prediction (cf. Figure 1 c). If there are no tiles assigned to a certrain part of a slide, the input for that part is a zero-tensor of the same dimension as other input tiles. This integrated CNN model with $k$ $\theta_e$'s and 1 $\theta_a$ is optimized end-to-end by the slide loss. In our experiments, cross entropy loss after softmax was used for tumor vs non-tumor classification (cf. Section 4.1), while binary cross entropy after sigmoid activation was used for multi-label prediction (cf. Section 4.2). The $k$ tile encoders $\theta_e$ share parameters during training.

Each slide can produce $p^k$ samples in the form of $\{x_1^i, ..., x_k^i\}$. To avoid overfitting, we construct only 10 samples per training slide, collect samples of all slides, randomly shuffle

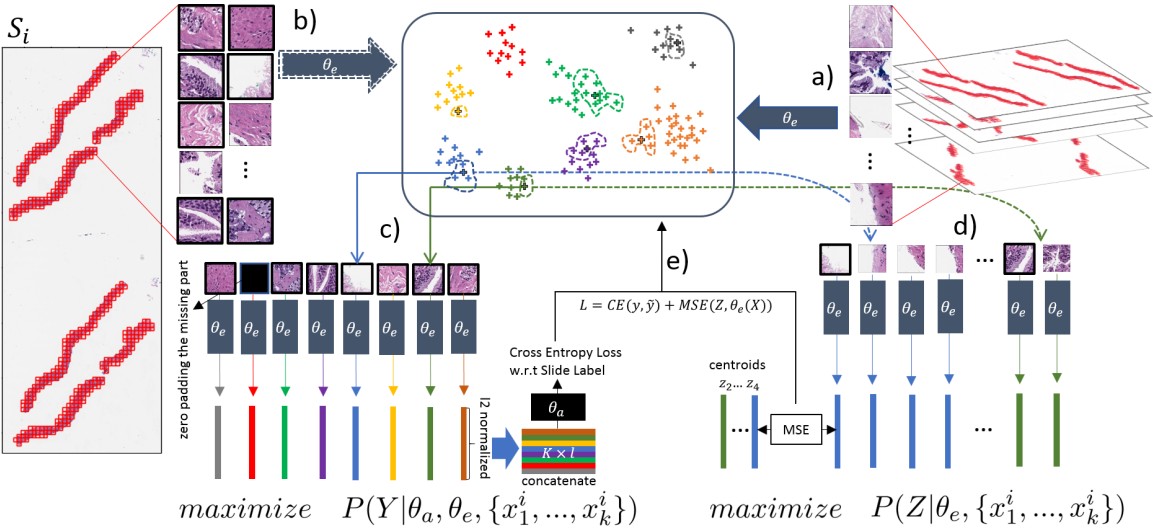

Figure 1: Overview of the proposed end-to-end part-learning (EPL) for WSI analysis. All tile encoder $\theta_e$'s share parameters. **a)** Manifold initialization: randomly assign the tiles of the whole training dataset to $k$ groups, randomly initialize $\theta_e$, and calculate the global centroids $\{z_1, ..., z_k\}$ of each group in feature space. **b)** Tiles of each training slide $S^i$ were mapped to the $k$ clusters in the manifold (dashed regions). **c)** Approximate the slide specific centroids $\{z_1^i, ..., z_k^i\}$ by their nearest tiles $\{x_1^i, ..., x_k^i\}$ (dark-outlined crosses). Feed the $k$ tiles to model for slide label prediction. **d)** Minimize the Euclidean distance between tiles of each part to their corresponding slide specific centroids in feature space. **e)** The 2 objectives are trained concurrently. The manifold and global centroids defined by $\theta_e$ change during training. Therefore tiles will be re-assigned to new centroids after each epoch.

them, and load a minibatch of size $n$ for each iteration of training. In the mean time, we also sample random tiles from the whole training dataset as a minibatch of size $2n$ for the centroid approximation learning (cf. Figure 1 d). The likelihood that the sampled tile features equal the corresponding centroids are maximized by MSE loss. The two losses are trained concurrently in each iteration.

### 3.4. Part Reassignment with Feature Attribution Estimation

As $\theta_e$ keeps being modified during training, the sampled tiles might no longer be good centroid approximations after certain iterations. Thus at the beginning of every epoch $t$, we calculate new global centroids $\{z_1, ..., z_k\}^t$ by averaging the new feature of each part of tiles assigned in the previous epoch $t - 1$: $z_k^t = 1/N \sum_{n=1}^{N} \theta_e^t(x_{k,n}^{t-1})$, then reassign tiles to the new centroids.

The part reassignment is based on the euclidean distance between tile features and centroids. However, the assignment at epoch $t$ should consider the *feature attribution* at $t - 1$, especially when the feature length $l$ is large. Otherwise, the important features learned from the previous epoch might be dominated by the other irrelevant ones in part formulation, which hinders the model learning convergence. We use the absolute value of the gradients w.r.t. the slide loss of the $k \times l$ features to estimate their contribution. In every epoch, the attribution of each feature of all samples are averaged, and used for euclidean distance calculation in a projected manifold for next epoch:

$$dist(\theta_e(x), z) = (\sum_i (a_i \theta_e(x)_i - a_i z_i)^2)^{0.5} \tag{4}$$

where $a_i$ is the attribution of feature i.

In this manner, the part reassignment, the centroid approximation by the nearest tiles, as well as the MSE loss for pushing the feature of these tiles to their centroids, will weigh the important features more than others.

### 3.5. Architecture and Inference

Each $\theta_e$ is a ResNet34 (He et al., 2016) whose last $fc$ layer is replaced by a $l_2$ normalization, i.e. the feature vector after global average pooling is $l_2$ normalized. Also, the feature length is reduced from 512 to $l$ by changing the output dimension of the last layer before global average pooling. $\theta_a$ is a one-layer $fc$ layer. The model is implemented with PyTorch (Paszke et al., 2017) and trained on a single Volta V100 GPU.

During inference stage, for good prediction, we use all tiles of each slide without any subsampling. According to the feature attribution and global centroids learned at training stage, the tiles are assigned to the $k$ parts and the nearest 1 tile to each slide specific centroid is selected. The $k$ tiles are fed through the model to output the slide prediction directly.

## 4. Experiments and Results

### 4.1. Clinical Grade Prostate and Basal Cell Carcinoma Classification

Recently, Campanella et al. (2019) used a novel MIL-RNN approach to define and achieve clinical grade WSI classification of prostate cancer and basal cell carcinoma (BCC). The

slides are labeled as positive if there are tumor regions on it, otherwise negative. This work provided us with a strong evaluation baseline of WSI binary classification. We obtained the same training, validation and test slides. For BCC data, there are 6900 training slides, 1487 validation slides and 1575 test slides. For prostate needle biopsies, there are 8521 trainig slides, 1827 validation slides and 1811 test slides. Each slide was tiled to patches of size $224 \times 224$ at $20\times$ magnification, which corresponds to 0.5 microns per pixel. Note that the tumor purity varies hugely among the slides, and the tumor regions on many of them span only a few tiles. Refer to (Campanella et al., 2019) for more details of the dataset.

We first evaluated the effect on convergence of utilizing feature attribution (cf. Section 3.4). We trained EPL with and without feature attribution (EPL-NA) on the prostate and BCC datasets for 2000 epochs at learning rate 0.1 with a decay factor of 0.1 after every 600 epochs, which took about 300 hours. The best model and hyper-parameters were chosen based the performance on the validation dataset. We used batch size $n = 32$, $k = 8$ clusters, and $p = 3$ nearest tiles for centroid approximation. For convergence study, we compared two feature lengths $l = 512$ and $l = 64$. The convergence curves are shown in Figure 2.

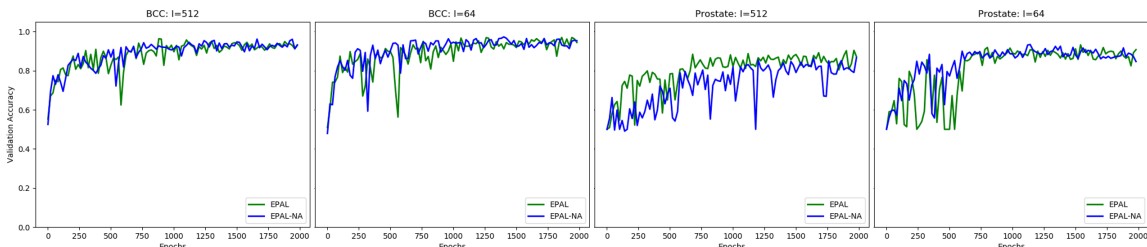

Figure 2: Convergence: EPL vs EPL with no feature attribution (EPL-NA). The prostate cancer classification training with $l = 512$ converges better with feature attribution.

The convergence of model learning on BCC dataset was good with or without feature attribution (cf. Figure 2 left). For prostate, the optimization converged generally better with smaller $l$ (cf. Figure 2 right). When $l$ is large (512), EPL converged better than EPL-NA. These observations are consistent with our theory that the convergence should benefit from feature attribution when $l$ is large (cf. Section 3.4), since there are more irrelevant noise features used in centroid approximation.

Since $l = 64$ gave better validation results, we used it in our final test stage. Table 1 shows the area under ROC curve of different tasks. The left panel compared EPL to the MIL and MIL-RNN approaches used by Campanella et al. (2019). It shows that EPL can reach comparable results to MIL. MIL-RNN learns another aggregation RNN on top of the learned features and achieved slightly better AUC. Note that Campanella et al. (2019) used quite strong tile-level supervision in their MIL training assuming that all tiles from non-cancer slides do not contain any tumor, i.e. $Z = Y$ (cf. Section 3.1). EPL can also be concurrently trained with these tile-level labels, which we surmise would further increase our performance. However the MIL assumption only holds for a very small portion of the tasks that EPL can be applied to. Besides, adding labels to tiles violates EPL's nature of

Table 1: Area under ROC curve for different classification tasks. Left: tumor vs non-tumor classification of prostate cancer and basal cell carcinoma. The results of MIL and MIL-RNN are from (Campanella et al., 2019). EPL has comparable clinical grade performance. Right: multi-label classification of lung cancer architectural subtypes. EPL-k1 represents the number of parts $k = 1$. Diverse parts are necessary for this multi-label classification task.

| Method | Cancer Classification | | Lung Cancer Architectural Subtyping | | | |
| | Prostate | BCC | Lepidic | Papillary | Solid | Micropapillary |
|---|---|---|---|---|---|---|
| MIL | 0.986 | 0.986 | - | - | - | - |
| MIL-RNN | 0.991 | 0.988 | - | - | - | - |
| EPL | 0.986 | 0.986 | 0.654 | 0.533 | 0.781 | 0.627 |
| EPL-NA | 0.984 | 0.987 | - | - | - | - |
| EPL-k1 | 0.734 | 0.930 | 0.585 | 0.518 | 0.648 | 0.530 |

end-to-end training. These extensions to EPL for optimal performance on certain tasks is beyond the scope of this paper. Nevertheless, our proposed method EPL achieved clinical grade performance for both prostate and basal cell carcinoma classification, with only 4 and 6 false negative slides (undetected cancer cases) out of the 1500+ test slides respectively. EPL-k1 represents $k = 1$. It lost the capability to combine information from diverse groups of tiles and resulted in much worse performance.

Beyond classification, EPL provides rich information of WSIs that is crucial for high quality clinical decision support. For any task learned by EPL, the importance of various tile groups can be estimated by averaging the feature attributions of the group. Figure 3 presents the part attributions side-by-side with the centroid approximation tiles used by EPL for prostate and BCC cancer classification. Each row corresponds to a part. Each column represents a test slide and the 8 tiles used for its classification. The heatmap bar represents the part attribution. 1's and 0's on top represents if it's cancer slide or not. A black square means that no tiles of this slide were assigned to this part.

The parts with high attribution for both tasks are basically the groups of cancer tiles (cf. Figure 3 from above, left: row 7,8; right: row 1,7,8). For these parts, the negative slides either contain no tiles belonging to them (black squares) or have non-tumor tiles similar in appearance to tumor, which should be from the sparse intersections between groups on the manifold. EPL combined different morphological subtypes of tumors for the final prediction. These subtypes, along with other identified parts, have potential biological meaning and clinical relevance, which is worthy to be studied carefully as an extension of this work. These tiles can also be mapped back to their original positions on WSIs (cf. Figure 5) for importance scoring of different regions over slide.

### 4.2. Multi-label Lung Cancer Architectural Subtyping

EPL's power of end-to-end learning of diverse features gives it potential for more complicated tasks of WSI assessment. As observed in prostate and BCC cancer classification

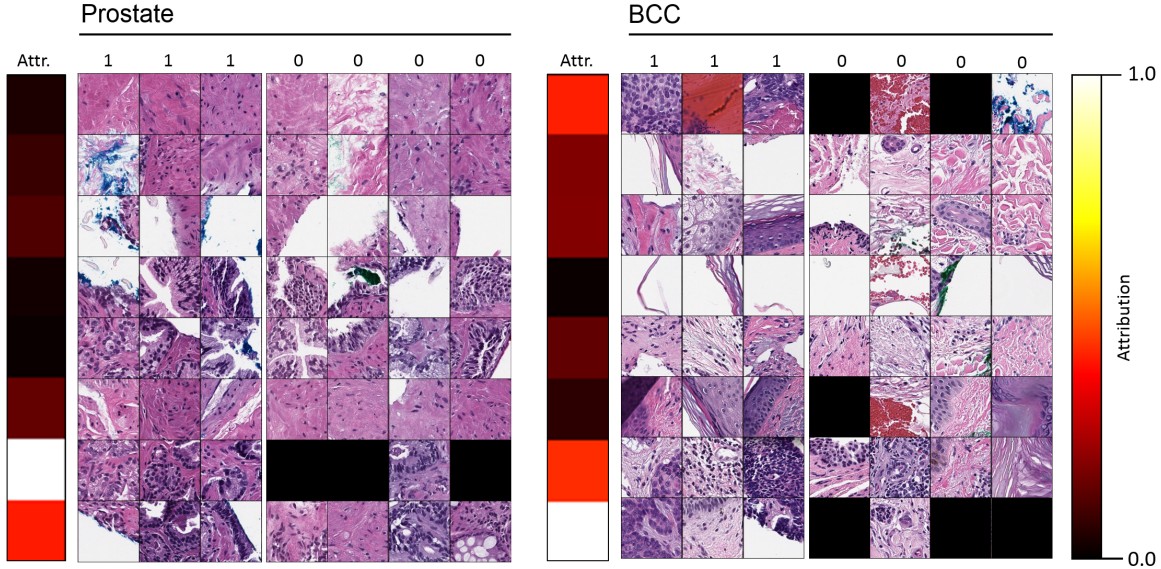

Figure 3: Part attribution and the centroid approximation tiles used by EPL for prostate and BCC cancer classification. Each row corresponds to a part. Each column represents a test slide and the 8 tiles used for its classification. The heatmap bar represents the part attribution. 1's and 0's on top means if it's cancer slide or not. A black square means that no tiles of this slide were assigned to this part. The parts with high attribution for both tasks are basically the groups of cancer tiles (from above, left: row 7,8; right: row 1,7,8). For these parts, the negative slides either contain no tiles belonging to them (black squares) or have non-tumor tiles similar in appearance to tumor. EPL combined different morphological subtypes of tumors for the final prediction.

(cf. Section 4.1), EPL has the potential to learn tumor subtypes, therefore we tested it on a curated dataset of weakly-supervised lung cancer architectural subtype prediction. The dataset contains 599 lung cancer primary resection slides from patients with lung adenocarinoma. Each slide was labeled with a vector of binary entries each representing the existence of an architectural subtype of {lipidic, papillary, solid, micropapillary} as indicated in the surgical pathology report. Since it is a smaller dataset, We did 5-fold cross-validation. We trained EPL on it for 1000 epochs with a learning rate 0.2 and a decay factor of 0.1 after every 400 epochs. Each epoch we used all training slides and 100 tiles per slide. Due to the small datasize, EPL began to overfit after 600 epochs, thus we selected the best model and hyper-parameters based on the performance on the validation dataset before reaching the point of overfitting. We used batch size $n = 32$, $k = 12$ clusters, feature lengths $l = 512$, and $p = 1$ nearest tiles for centroid approximation.

The area under ROC curve based on the prediction score of each subtype is shown in Table 1 right panel. For this weakly-supervised multi-label prediction task, MIL can't be applied without substantial adaptation. Among the 4 architectural subtypes, EPL predicted solid tumor the best, while learning the exsistence of papillary imposed hardness. When $k = 1$, the capability of EPL degraded due to the loss of feature diversity. We found that the groups learned by EPL were formed such that some of them directly corresponded to the four subtypes used for training quite well (cf. Figure 4 row 6,9,10,11). Each row in Figure 4 is the 20 nearest tiles of the whole validation dataset to the learned global centroids. Note that although not used in the slide label, the clinically relevant acinar subtype was enriched in part 7 (cf. Figure 4 row 7). Given these observations, we surmise that gathering more data would help preventing the overfitting and achieving better results.

## 5. Discussion

As a general weakly-supervised WSI prediction algorithm, EPL was built upon the least assumptions comparing to MIL approaches, and thus theoretically applicable to a plethora of tasks. This, however, implies that EPL might require large datasets to be most successful. To achieve better accuracy on smaller datasets, EPL can be easily combined with supervision for tile-level proxy tasks. For example, EPL applied to the lung subtyping task described in this paper can be concurrently trained with subtype labels on a small number of tiles. Another extension of EPL is it's application on survival regression. As many other works have shown (Muhammad et al., 2019; Yao et al., 2019; Zhu et al., 2017b), modeling WSIs as clusters of tiles is uniquely powerful for survival analysis. Implementing EPL on survival analysis differs from them by not only learning tile groups, but also forming the groups based on slide level supervision. With the simplicity and power of its end-to-end structure, we suggest that EPL can be the backbone or framework, based on which, many innovative and efficient models can be developed for WSI assessment.

## Acknowlegements

This work was supported by the Warren Alpert Foundation Center for Digital and Computational Pathology at Memorial Sloan Kettering Cancer Center, the NIH/NCI Cancer

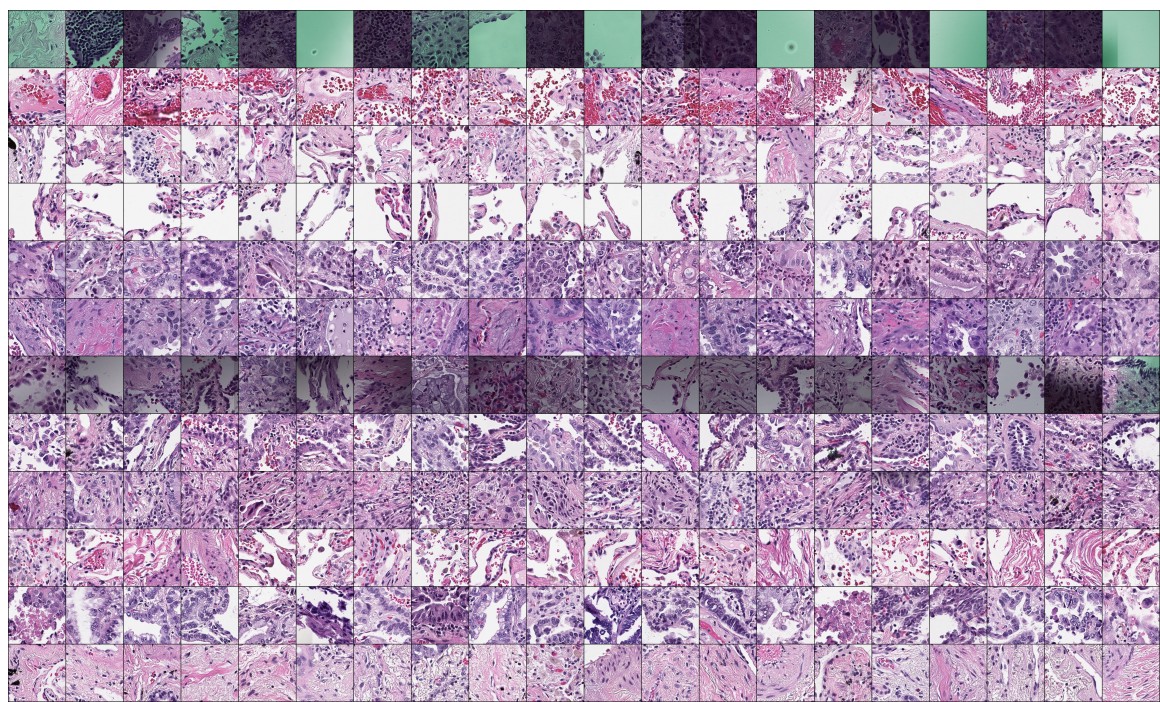

Figure 4: From top to bottom: 1. Green ink. 2. Red blood cells in blood vessels near alveolar spaces. 3. Macrophages in alveolar spaces, often with hemosiderin in the macrophages. 4. Normal alveolar wall. 5. Cancer enriched for micropapillary subtype. 6. Cancer enriched for acinar subtype. 7. Black ink. 8. Cancer enriched for lepidic subtype. 9. Cancer enriched for high grade morphology, solid like. 10. Blood vessel and alveolar wall with sparse cells in spaces. 11. Cancer enriched for papillary subtype. 12. Stroma.

Center Support Grant P30 CA008748, Weill Cornell Graduate School of Medical Sciences and the Tri-I Computational Biology and Medicine Program.

T.J.F. is the chief scientific officer, co-founders and equity holders of Paige.AI. C.X. and T.J.F. have intellectual property interests relevant to the work that is the subject of this paper. MSK has financial interests in Paige.AI. and intellectual property interests relevant to the work that is the subject of this paper.

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

**Appendix A. Tissue Type Localization and Region Importance Scoring**

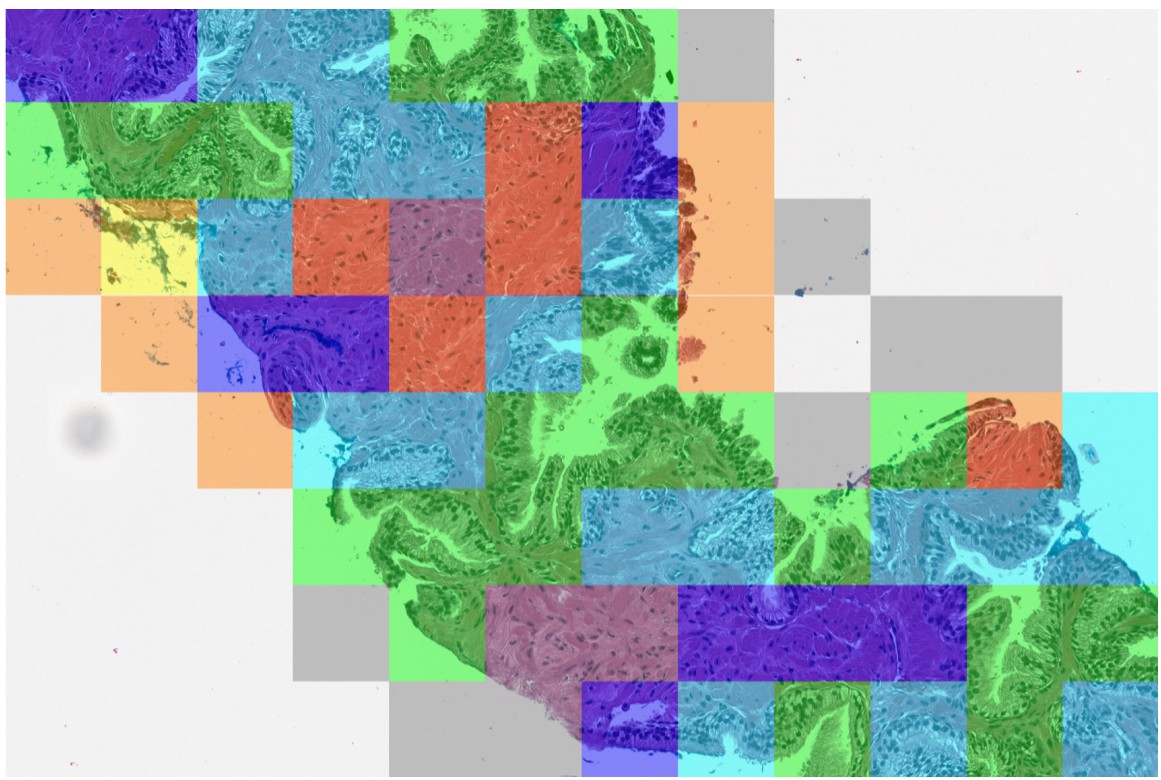

Figure 5: Mapping of the learned parts back onto a prostate needle biopsy. Each part is linked to an importance score (attribution). Colors were randomly chosen and irrelevant to attribution. The tiles to be highlighted for clinical decision support can be filtered by their distance to the part centroids.

