# OpenReview forum: "Beyond Classification: Whole Slide Tissue Histopathology Analysis By End-To-End Part Learning"
_MIDL.io/2020/Conference — MIDL 2020_

### Official Review · AnonReviewer4 · 2020-03-06
**why an acinar pattern disappeared?**

**Rating:** 4
**Confidence:** 4
**Recommendation:** Oral

**Summary:**

Presented work is interesting, innovative and well described. The authors proposed a two-stage method for WSIs classification that that is able to learn diverse features from tiles. The authors performed experiments on large datasets (thousand slides used for training) for three independent tasks.  Both, developed methods and presented results are interesting for a science community.

**Strengths:**

Strengths:
- experiments on large datasets (thousand slides used for training) for three independent tasks
- innovative method
- a method is well described
- results for prostate cancer and basal cell carcinoma were compared with results available in the literature


**Weaknesses:**

The paper has one significant issue, that is an acinar pattern in lung classification that disappeared. The adenocarcinoma has five main patterns: solid, micropapillary, papillary, lepidic and acinar. In figure 4 acinar pattern is presented in row 6. However, in the description and results, the acinar pattern is not mentioned. As a result, it is confused if an acinar was a subtype that occurred in the training/test dataset and was detected and why this pattern is not evaluated in the table. It looks as an acinar pas a part of the training dataset but was excluded from the evaluation. Why?  As well is it not clear from where came lung cancer dataset includes 599 WSIs (from single or many medical centers)?

**Detailed Comments:**

Missing comparison/ references to the current state-of-the-art lung adenocarcinoma classification/ segmentation results.

**Justification Of Rating:**

The authors presented a clear goal and an innovative method. The proposed solution was developed based on large datasets and evaluated on three independent tasks.  The main issue that should be explained is the disappearing acinar pattern.

**Paper Type:**

both

**Questions To Address In The Rebuttal:**

It should be clarified why the acinar pattern is not evaluated but occur in the dataset (fig. 4, row 6). The lung dataset should be described, from where it was collected, who prepared annotations (pathologist? students?).

**Special Issue:**

yes

---

> ### Author Response · Authors · 2020-03-27
> **Response to AnonReviewer4**
>
> 1.) Acinar subtype:
> The acinar subtype was originally also labeled for each slide, however we found the training became better if we exclude acinar subtypes from the label vector. Although it's not part of the supervision during training, we found that the acinar tiles were also grouped together by EPL in the end.
>
> 2.) Lung dataset:
> The lung dataset were collected by our own institution. The labels were created based on manual evaluation on the pathology reports and one representative slide for each part was collected.

---

### Official Review · AnonReviewer2 · 2020-03-10
**A contribution to pathology-specific deep learning models, but lacks compelling reason for adoption**

**Rating:** 3
**Confidence:** 4
**Recommendation:** Poster

**Summary:**

This paper demonstrates a novel deep learning architecture for whole slide images (WSIs). To reduce the computational burden of processing WSIs, all tiles from the WSI are encoded into the feature space and clustered into groups. The model weights are then learned for each cluster. In inference, a single tile from each group is used to make a decision. This greatly reduces the number of tiles that need to be processed to render a decision. This architecture is compared against a well-performing existing model and found to have similar performance.

**Strengths:**

The novel architecture described in this manuscript seems to be based on solid mathematical and machine learning foundations, and is adequately described here.

The use of an external, publicly available dataset is a strength.

The grouping of tiles into clusters lends some interpretability to the model, as shown in Figures 3, 4, and 5.



**Weaknesses:**

The need for this model is unclear. WSIs are difficult to process, but this model appears to perform exactly as well as the MIL model it is compared to and worse than MIL-RNN, weakening the motivation for adoption of this new architecture.

Redesigning Figure 1, describing the model, would make it easier to understand this paper.

**Detailed Comments:**

k  is used to refer to both the number of groups and number of tiles. But it does not appear that each group consists of a single tile.

Figure 1 is confusing. The natural reading order takes the reader on the path S_i -> b ->  a -> c -> e -> d.

Figure 1 (a) describes allocating the entire dataset to k groups, and Figure 1 (b) refers to mapping the training set within these clusters. This suggests the clustering depends on the test set, though section 3.3 clarifies, it may be worthwhile to make this clear in the caption as well.

What is the dimensionality of the mapping space?

There are a number of typos in this manuscript, “certrain”, “out experiments”, “Conclusively, Weakly”

The number of patients in the dataset, number of slides per patient, and number of positive and negative slides should be given, as it is important enough to be restated in this paper.

 Figure 3 and Figure 4 are interesting and informative, however the caption from Figure 3 appears to be directly copied from the text and could be improved by changes such as replacing the 1’s and 0’s with multi-column spanning labels “cancer slides” and “non-cancer slides”.


**Justification Of Rating:**

This paper has no glaring flaws and there is a need to develop deep learning architectures specific to digital pathology. The model design is reasonable and suited to the clinical problem. However there does not seem to be a compelling reason to adopt this architecture over other schemes.

**Paper Type:**

methodological development

**Questions To Address In The Rebuttal:**

What is the primary motivation for this work? The authors offer a few reasons, including less laborious annotating, faster processing, learning tile groupings, and applicability to multiple clinical tasks, but it is not clear what the main focus is. What is the most pressing unmet need that this paper solves?

It is possible that the lung cancer subtyping experiment could demonstrate that this model outperforms existing gold standards, but that experiment is limited by the use of cross validation and the intervention to stop model training before overfitting. The use of cross validation and selection of the best model and hyper parameters based on the validation set suggests that the model that performed best on the validation set was selected, meaning the reported metrics are likely overoptimistic of performance on an independent validation set.

**Special Issue:**

no

---

> ### Author Response · Authors · 2020-03-27
> **Response to AnonReviewer2**
>
> Please refer to our general response to the reviewers' comments for most of the points need to be discussed here. We believe remaking Figure 1 would polish EPL and clarify it's unique contribution to the field of WSI assessment.
>
> Lung subtyping and dataset:
> We chose the best model before overfitting, which performed both good on training and validation datasets. All of our datasets are balanced and one patient has only one slide. We will report the number of positive and negative slides in the dataset in the final version.

---

### Official Review · AnonReviewer3 · 2020-03-13
**End-to-end part learning approach**

**Rating:** 3
**Confidence:** 4

**Summary:**

This paper presents an end-to-end part learning (EPL) approach as an alternative to conventional two-stage approaches (tile encoder and tile aggregation), generally used for analyzing histopathology whole-slide images (WSIs). Each WSI is clustered into k groups, the proposed model jointly learns the class label for image patches and clusters global centroids. This study is performed on 3 datasets, including prostate cancer, basal cell carcinoma, and lung cancer. The lung cancer dataset contains multi-class labels and the other two datasets treated as binary class problems.

**Strengths:**

Overall, the manuscript is well-written and addressing a relevant problem by proposing an interesting end-to-end part learning method. The performance of the model is validated on 3 datasets from different indications.

**Weaknesses:**

Below are some minor/major comments
- Table 1 lacks comparative analysis (especially for the multi-class problem) between conventional two-stage classification approaches and the proposed approach which makes it difficult to quantify how well the proposed approach is performing. The performance with an existing MIL approach is comparable (nearly same).
- The proposed approach is heavily based on some of the hyper-parameters, for instance, k, there is no empirical evidence that how one should select k and how the performance of the model will vary by changing k. Besides, the value of k is different for different datasets. Another parameter is p for centroid approximation, again, no ablation experiments are performed to validate the robustness of the selected value for p.
-  In section 3.3, it is stated that the whole training data were randomly split into k groups. Is this the best way to split data into k clusters (centroid initialization methods)?
- During inference, is the proposed approach less computational expensive then MIL or conventional two-stage approaches? It would be worth reporting the run-time (e.g in seconds) or computational complexity or number of learnable parameters.
- In Table 1, the last row where k=1, with the BCC dataset the model performance is 0.93, if during inference only k tiles are fed through the model to output the slide prediction then it would be of interest to show that one patch that classifies the WSI label.
- In mathematical notations, some of the variables are not defined properly like N, the relationship between a slide S and X and xi is not clear.

Some very minor comments,
- The following sentence needs a revision: binary cross entropy after sigmoid activation was used for multi-label prediction.
- Title of the paper: Classfication -> Classification.
-  In out experiments -> In our experiments.

**Justification Of Rating:**

The paper presents an interesting end-to-end part learning approach by using 3 datasets but some of the key comparative and ablation experiments are missing. There are some inconsistencies in mathematical notations.

**Paper Type:**

methodological development

**Special Issue:**

yes

---

> ### Author Response · Authors · 2020-03-27
> **Response to AnonReviewer3**
>
> We thank Reviewer 3 for the suggestions on ablation studies.
>
> 1.) Comparing to MIL:
> Please first refer to our general response to the reviewers' comments. The MIL approach (Campanella et al., 2019) on the prostate and BCC datasets is the state-of-the-art method reported in the literature on large scale evaluation. Other works mentioned in our related work section adopted similar approaches but benchmarked against small curated datasets. We regard the method proposed by Campanella et al. (2019) as a strong baseline and decided to benchmark EPL against the same large scale datasets.
>
> 2.) Hyperparameters:
> We experimented with different values of the parameters $k$ and $p$ during the development. The empirical results showed that EPL is not sensitive to $k$ when $k>=6$, however if $k$ is too small (we tried 4 and 1), the performance degraded on BCC and especially prostate dataset. We surmise the reason is that there are natural major types of tiles, such as stroma and tumor, therefore pushing these tiles together in the feature space imposes hardness on learning. When $k=1$, it means that in inference stage EPL only look at one tile per slide; for the BCC and prostate cancer classification, that one tile will be very likely what a traditional MIL-max approach would be looking at. We tried $p=1,2,3$, which seemed quite robust on the BCC and prostate datasets. On the lung multi-label subtyping task, we found $p=1$ worked the best. Large $p$ means worse centroid approximation and introduces noise to the slide label learning based on the part representation tiles. We would like to add these discussions in the final version.
>
> 3.) Mathematical expression:
> We will re-examine our mathematical expression and correct minor inconsistencies if any.

---

### Official Review · AnonReviewer1 · 2020-03-16
**Novel way of training CNN end-to-end to classify histopathology images**

**Rating:** 4
**Confidence:** 5
**Recommendation:** Oral

**Summary:**

This paper introduces a method to learn a system based on convolutional networks for classification of whole-slide images.
The approach assumes that parts of the whole-slide image can be grouped based on unsupervised cluster analysis, and therefore only representative patches close to cluster centroids are involved during training, making the system trainable end-to-end.
The method is validated on two applications and three datasets: cancer detection in prostate and basal cell carcinoma, and lung adenocarcinoma growth pattern classification.
Results on the first application are close to a recently presented state of the art system on the same test set; no comparison on the second application is presented.

**Strengths:**

* valid alternative to existing solutions for weak supervised learning in histopathology image classification
* method trained and validated validated on fairly large datasets
* efficient use of computational resources by using cluster analysis

**Weaknesses:**

* results close to the state of the art for the detection method, but slightly worse (compared to MIL-RNN). Authors should explain what is the added value of using the proposed method in this application. If its strenght is in computational efficiency, then a specific comparison in this direction should be presented.
* Authors state that this method could be used to predict survival but no experiment is reported. The title of the paper is "beyond classification", but in the end only classification is shown.
* Figure 1 is quite confusing, despite the description in the caption, it is quite hard to follow the workflow of the method. What is training set? What is test set? Where is the training of model parameters happening? Consider rearranging the order of the components.
* From Figure 1, it seems that clusters are made using patches from the full dataset, and later patches from the training set are mentioned. Does this mean that clusters are defined using patches from images of the validation and the test set as well? This should be clarified.
* It is not clear what the contribution of Figure 5 is, as colors are not explained, and tissue under colored patches is difficult to recongize.

**Justification Of Rating:**

This paper introduces a method to train CNN end-to-end for whole-slide image classification.
It presents an approach that is novel compared to existing work in the field and therefore represents an important contribution.
The paper is well written and relatively easy to follow.
Results are in line with the state of the art.
More datasets should be analyized, also aiming at predicitng survival.

**Paper Type:**

methodological development

**Special Issue:**

yes

---

> ### Author Response · Authors · 2020-03-27
> **Response to AnonReviewer1**
>
> We are grateful to Reviewer 1 for the positive and constructive comments. We want to address the following points in detail:
>
> 1.) Comparing to MIL:
> Please first refer to our general response to the reviewers' comments. A detailed discussion about comparing to MIL-RNN was in paragraph 4, Section 4.1. As an extension, EPL can also be combined with an RNN on top of the clusters and attribution to clusters it learned, but it's beyond the scope of this paper that focuses on presenting the end-to-end nature of our model.
>
> Survival analysis is theoretically a natural application of EPL, as the unsupervised clustering methods have shown state-of-the-art performance on those tasks (Section 5). We will move some of the discussions in Section 5 to introduction and related work sections.
>
> 2.) Algorithm:
> The dataset was split to training, validation and test sets at patient level. The whole training of EPL was based on training set. We are using all tiles of the training set (not the "whole dataset") to define the manifold. This expression error will be corrected in the final version with an updated Figure 1.

---

### Author Response · Authors · 2020-03-27
**General response to the reviewers’ comments**

Here we want to address some common discussions that multiple reviewers brought up.

1.) The unmet need in the field that EPL solves:
We are very inspired by the reviewers' comments on how we should improve conveying the contribution of EPL to the field of WSI assessment. Although we believed enough discussion was presented in our introduction and related work, and especially in Section 3.1, those discussions might biased too much to mathematical explanation. We would like to reformulate some of the descriptions and adding more application examples in these discussions here and in the final version of the paper.

Our proposed end-to-end part learning (EPL) model was designed to solve tasks that are harder than cancer classification. To predict whether a WSI contains tumor, a model only needs to classify each tile to tumor or non-tumor, and the slide is tumorous if at least one tile is tumor (Z=Y; Section 3.1). MIL approaches work perfectly in this scenario, but also only applicable to problems with this assumption. For a more complicated task, such as the multi-label (not multi-class) prediction of the existence of lung cancer subtypes on a WSI (Section 4.2), no single tile can represent the label of the slide as a vector (Z$\neq$Y), for example $<0,1,1,1>$ where "1" means the subtype exists. MIL approaches can't be applied to this problem without substantial adaptation, thus not compared in Table 1. Our logic is to first benchmark EPL against an established standard on large scale datasets on traditional tasks, then showed that it has the potential to solve a new multi-label problem.

We are now also using EPL for survival regression and prediction of patients' response to immunotheropy based on WSI's and start to see some promising results. In these two scenarios, the slide label is related to the tile features in a even more complicated way. Our MIL implementation on these tasks completely failed. EPL solves WSI assessment by an integrated end-to-end architecture, can theoretically learn to combine diverse information over the whole slide for any learnable tasks.

2.) Algorithm illustration with Figure 1:
We will redesign Figure 1 to better describe the algorithm in the final version. We also consider adding an "EPL Algorithm" pseudo code. Here we want to emphasize again that EPL is optimized only on training dataset, parameters are determined by validation dataset, and the results are on test dataset which was never touched before. Also, EPL learns the whole integrated model based only on slide label. Therefore it can be regarded as "supervised cluster learning". This is in contrast to the other clustering approaches that learns the encoder based on unsupervised contraint losses, such as tile reconstruction or contrastive loss.

---

### Meta-Review · Area_Chair1 · 2020-04-07
**MetaReview of Paper305 by AreaChair1**

**Rating:** 4
**Recommendation For Accepted Papers:** Oral

**Metareview:**

All reviewers indicate acceptance and highlight that the method is interesting and solves a relevant problem. The authors show their method's performance on several relevant tasks and compare it to a baseline. As such, I also strongly recommend acceptance.

**Paper Type:**

methodological development

**Special Issue:**

yes

---

### Decision · Program_Chairs · 2020-04-11

Accept